# Short-term associations of diarrhoeal diseases in children with temperature and precipitation in seven low- and middle-income countries from Sub-Saharan Africa and South Asia in the Global Enteric Multicenter Study

Nasif Hossain [1,2]*, Lina Madaniyazi[1], Chris Fook Sheng Ng[1,2], Dilruba Nasrin[3,4], Xerxes Tesoro Seposo[5], Paul L. C. Chua[2], Rui Pan[2], Abu Syed Golam Faruque[6], Masahiro Hashizume[1,2]

1 Department of Global Health, School of Tropical Medicine and Global Health, Nagasaki University, Nagasaki, Japan, 2 Department of Global Health Policy, School of International Health, Graduate School of Medicine, The University of Tokyo, Tokyo, Japan, 3 Center for Vaccine Development and Global Health, University of Maryland School of Medicine, Baltimore, Maryland, United States of America, 4 Department of Medicine, University of Maryland School of Medicine, Baltimore, Maryland, United States of America, 5 Department of Hygiene, Graduate School of Medicine, Hokkaido University, Sapporo, Japan, 6 Nutrition and Clinical Services Division, International Centre for Diarrhoeal Disease Research, Bangladesh(icddr,b), Mohakhali, Dhaka, Bangladesh

* nasif.stat.iu@gmail.com

## Abstract

### Background

Diarrhoeal diseases cause a heavy burden in developing countries. Although studies have described the seasonality of diarrhoeal diseases, the association of weather variables with diarrhoeal diseases has not been well characterized in resource-limited settings where the burden remains high. We examined short-term associations between ambient temperature, precipitation and hospital visits due to diarrhoea among children in seven low- and middle-income countries.

### Methodology

Hospital visits due to diarrhoeal diseases under 5 years old were collected from seven sites in The Gambia, Mali, Mozambique, Kenya, India, Bangladesh, and Pakistan via the Global Enteric Multicenter Study from December 2007 to March 2011. Daily weather data during the same period were downloaded from the ERA5-Land. We fitted time-series regression models to examine the relationships of daily diarrhoea cases with daily ambient temperature and precipitation. Then, we used meta-analytic tools to examine the heterogeneity between the site-specific estimates.

### Principal findings

The cumulative relative risk (RR) of diarrhoea for temperature exposure (95th percentile vs. 1st percentile) ranged from 0.24 to 8.07, with Mozambique and Bangladesh showing positive

**Data Availability Statement:** The data underlying the results presented in the study are available for download from https://github.com/nasifenvepi/Temperature-and-precipitation-association-with-child-diarrhoea.

**Funding:** This work was supported by the Japanese Government (Monbukagakusho) Scholarship from the Ministry of Education, Culture Sport, and Technology (MEXT-192202 [NH]) and the Environment Research and Technology Development Fund (JPMEERF23S21120[MH]) of the Environmental Restoration and Conservation Agency provided by Ministry of the Environment of Japan. This work was also partly supported by the Bill & Melinda Gates Foundation, Grant no. NV-002050 [ASGF,DN]. The funders had no role in study design, data collection, analysis, publication decisions, or manuscript preparation.

**Competing interests:** The authors have declared that no competing interests exist.

associations, while Mali and Pakistan showed negative associations. The RR for precipitation (95th percentile vs. 1st percentile) ranged from 0.77 to 1.55, with Mali and India showing positive associations, while the only negative association was observed in Pakistan. Meta-analysis showed substantial heterogeneity in the association between temperature–diarrhoea and precipitation–diarrhoea across sites, with $I^2$ of 84.2% and 67.5%, respectively.

## Conclusions

Child diarrhoea and weather factors have diverse and complex associations across South Asia and Sub-Saharan Africa. Diarrhoeal surveillance system settings should be conceptualized based on the observed pattern of climate change in these locations.

## Author summary

Diarrhoeal diseases remain a significant public health concern, particularly in low- and middle-income nations, and understanding the environmental factors that influence their occurrence is crucial for effective prevention and management strategies. Here, we study the weather factors, such as temperature and precipitation, that could influence the prevalence of diarrhoeal diseases in children. We found that higher temperatures were associated with an increased risk of diarrhoea in some regions. Increased precipitation was associated with a higher incidence of diarrhoea in some sites. In certain sites, diarrhoeal cases decreased with rising temperatures and precipitation. These findings offer insights into the climate and geographic patterns of childhood diarrhoea, which are essential for developing targeted and efficient public health interventions. Recognizing the significant role of weather-related variables in driving child diarrhoea, it is imperative to identify the conditions associated with these patterns to enhance our understanding of the complex interplay between environmental factors and disease prevalence.

## Introduction

Diarrhoeal disease is a major cause of death in children under 5 years old, claiming the lives of approximately 443,832 children every year [1]. Although the number of episodes of childhood diarrhoea is declining globally, the prevalence of diarrhoeal disease remains high in many resource-poor settings where infants and children are at risk of death and other infectious diseases such as malaria and pneumonia [1,2]. In South Asian regions, diarrhoea-related morbidity remains alarmingly high, with more than 2000 children dying daily from diarrhoeal diseases, far more than the number dying from HIV/AIDS, malaria, and measles combined [3]. In Sub-Saharan Africa, diarrhoea remains a leading cause of death in children under the age of 5 years, accounting for approximately 750,000 of the total 4.3 million deaths among African children before their fourth birthday [4].

The Global Enteric Multicenter Study (GEMS) is the largest, most comprehensive study of childhood diarrhoeal diseases ever conducted in developing country settings [5]. The study sites exhibit diverse climates, highlighting environmental variability. Mirzapur (Bangladesh) and Kolkata (India) share tropical monsoon and wet-and-dry climates, respectively, with hot summers, heavy monsoon rains, and cooler winters, peaking in precipitation in July and August. Karachi (Pakistan), with a hot desert climate, has hot, dry summers, a brief monsoon

season, and mild winters, receiving low annual precipitation mainly during the monsoon. Nairobi's (Kenya) subtropical highland climate features mild year-round temperatures and two rainy seasons. Banjul (The Gambia) and Bamako (Mali) have tropical climates with distinct wet and dry seasons, peaking in precipitation in August. Manhiça (Mozambique), with a tropical savanna climate, experiences a wet season from November to April and a dry season from May to October, peaking between January and March.

A complex set of weather factors is involved in driving the prevalence of diarrhoeal diseases. With climate change, there is concern that the incidence and severity of diarrhoea may increase owing to altered patterns of temperature and precipitation, as well as extreme weather events [6,7]. Temperature and precipitation variations generally impact diarrhoea prevalence through their effects on the survival and growth of various bacteria, protozoa, and viruses that cause infections, of which diarrhoea is a symptom [8]. Several time-series studies have investigated the relationship between specific climatological variables and all-cause diarrhoea in particular regions, sites, and countries, including the relationship with temperature in Peru [9], Vietnam [10], China [11], Australia [12] and Fiji [13] and that with precipitation in Mozambique [14], Ecuador [15], the United States [16], and Rwanda [17]. However, most studies have focused on either temperature or precipitation, and only a few investigations have considered both the effects of temperature and precipitation on diarrhoea cases [18,19]. Additionally, these studies have been limited to analysing weekly or monthly diarrhoea counts, and none have precisely quantified the association between temperature and precipitation with daily diarrhoea counts in multiple settings and diverse contexts. The objective of this study was to investigate the short-term relationship of childhood diarrhoea with temperature and precipitation in seven low- and middle-income countries by age group. The data from the seven GEMS sites provide good coverage of children populations in sub-Saharan Africa and South Asia where approximately 90% of diarrheal mortality has been documented [20], where rising temperatures and erratic precipitation have been documented in the last few decades [21,22].

## Methods

### Data collection

Our study included data from both hospitalized and non-hospitalized under-5 child diarrhea patients who visited the GEMS-nominated hospital, a secondary data source between December 1, 2007 and March 3, 2011. We included patients aged 0–59 months who met the case definition for diarrhoea (≥3 abnormally loose stools within 24 hours) and were referred to as having all-cause diarrhoea (ACD). Using dates on patients' individual records, we generated a site-by-site daily time-series dataset. In the absence of an individual record on a given day, we assumed no patients on that day [23]. Our study's total number of ACD cases was 66,056 children, as outlined in Table 1. The ACD data used for this study were different from the original GEMS data (matched case-control). The GEMS study was mainly a case-control study of acute diarrhoea in children 0–59 months. Specifically, the GEMS study enrolled both case children (symptomatic moderate to severe diarrhoea) and matched control children (asymptomatic) who met the eligible criteria for participation. Moderate to severe diarrhoea (MSD) was selected from all-cause diarrhoea patients if they met pre-defined inclusion criteria [5]. The first 8–9 patients were enrolled as a case per fortnight in each three-age stratum (0–11 months, 12–23 months, and 24–59 months) at the Sentinel Health Centers (SHCs). During the study period, GEMS was able to recruit 9,439 children with MSD and 13,128 matched controls.

This study included seven study sites representing a range of developing countries' child health indicators and mixers of urban, peri-urban, and rural settings. Where Bamako (Mali), Kolkata-W. Bengal (India) urban areas, Mirzapur (Bangladesh) and Karachi (Pakistan) are

Table 1. Summary Statistics of Daily Study Variables in Seven Sites, 2007–2011.

| Study Site | All-cause diarrhoea, n (%) | | | | | |
|---|---|---|---|---|---|---|
| | Age 0–11 months | Age 12–23 months | Age 24–59 months | Total | Average Temperature ˚C (range) | Total Precipitation mm (range) |
| The Gambia | 3047 (40%) | 3415 (44%) | 1226 (16%) | 7688 | 28.8 (21.1, 37.4) | 5.48 (0.0, 305.7) |
| Mali | 4315 (47%) | 3132 (34%) | 1791 (19%) | 9238 | 27.5 (17.4, 37.7) | 6.90 (0.0, 135.7) |
| Mozambique | 5488 (42%) | 4914 (38%) | 2583 (20%) | 12985 | 23.5 (16.0, 32.5) | 4.89 (0.0, 145.7) |
| Kenya | 1148 (48%) | 706 (30%) | 530 (22%) | 2384 | 22.0 (18.8, 25.3) | 13.4 (0.0, 118.5) |
| India | 1967 (41%) | 1716 (36%) | 1085 (23%) | 4768 | 26.4 (15.9, 33.3) | 11.4 (0.0, 209.8) |
| Bangladesh | 3410 (47%) | 2505 (35%) | 1327 (18%) | 7242 | 25.5 (15.2, 31.9) | 15.6 (0.0, 326.2) |
| Pakistan | 9017 (41%) | 6099 (28%) | 6635 (31%) | 21751 | 26.2 (15.9, 34.1) | 1.5 (0.0, 132.5) |

peri-urban areas, and Basse (The Gambia), Manhica (Mozambique), and Nyanza Province (Kenya) are rural areas.

Hourly weather data over the study period were from the European Centre for Medium-Range Weather Forecast Reanalysis Version 5-Land (ERA5-Land) and downloaded from the Copernicus Climate Data Store (https://cds.climate.copernicus.eu/#!/home). The dataset from ERA5-Land provides high spatial and temporal resolution, allowing for a 9 km grid at hourly time steps; vertical coverage is from 2 metres above the surface level. These datasets include hourly 2-metre ambient temperature-also known as outdoor temperature (degree Kelvin), hourly dew point temperature (degree Kelvin), and hourly precipitation (metres). The hourly time-series variables were converted into local time zones (S1 Table) from Coordinated Universal Time (UTC)±00:00. We calculated the daily averages for temperature and dew point temperature (converted into ˚C) and aggregated daily total precipitation in millimetres.

The weather data were matched with the daily patient count for all-cause diarrhoea at each site, based on location (nearest ERA5-land grid square centre of each study site) and date. Our initial inspection of the data suggested highly skewed precipitation data (S2 Fig and S1 Table). Thus, we calculated the 21-day rolling sum of daily total precipitation, referred to as cumulative precipitation, to consider the broad diarrhoeal disease transmission processes [7,24]. We determined a 21-day period prior to the hospital visit as our exposure window. We assume that considering precipitation occurrences, transmission of pathogens, subsequent onset of diarrhoea infections, and hospital visits, these events are likely to unfold over approximately 21 days [25]. Relative humidity was calculated as a percentage using temperature and dew point temperature [23].

We also included site-specific indicators. GDP per capita is sourced from Gridded Gross Domestic Product (per capita) data from Dryad. Population density is sourced from Gridded Population of the World (GPW) using UN WPP Adjusted Population Density. Poverty levels are derived from the Global Gridded Relative Deprivation Index (GRDI) and Urbanization-the gridded global urban land expansion product at a 1-km resolution are derived from PANGAEA. Other WASH and health-related indicators such as treated water, the primary source of drinking water, handwashing practices, the prevalence of stunting, viruses, bacteria, protozoa, moderate to severe diarrhoea (MSD) among under-five children, and percentage of treated Oral Rehydration Solutions (ORS) are sourced from the GEMS case-control study (S3 Table) [5].

## Statistical analysis

The impact of ambient temperature and precipitation was assessed by a two-stage modelling framework. In the first stage, we assessed the association of diarrhoea with temperature and

precipitation for each site by using time series regression analysis. In the second stage, we used meta-analytic tools to examine the heterogeneity between the site-specific estimates.

We assume the outcome variable daily diarrhoea cases followed quasi-Poisson distribution because of overdispersion. To assess the associations of temperature and precipitation with daily diarrhoea visits in children, we fitted a time-series regression model with quasi-Poisson distribution by using a distributed lag nonlinear model (DLNM) for each country. DLNMs are specified by the definition of a cross-basis, a bi-dimensional functional space describing at the same time the dependency along the range of the temperature and in its lag dimension [26].

$$Y \sim Poisson(u_t)$$

$$Log[E(Y_t)] = \beta_0 + CB_t + ns(rs\_preci_t) + ns(time_t) + DOW_t + Holiday_t$$

Where $Y_t$ is the number of diarrhoea cases on day $t$ for the diarrhoea outcome; $\beta_0$ is the intercept; $CB$ is the cross-basis function of temperature (smoothed using a natural cubic spline [$ns$] with 3 degrees of freedom [df]) and its lags (21-day lags with 3 equally spaced knots in the log scale). The cross-basis function allows us to fit the non-linear exposure-response association and delayed effects of temperature simultaneously, by building a grid of predictors for each lag and for each temperature exposure levels. We obtained the overall cumulative effects predicted from the daily temperature on all-cause diarrhoea over 21 days, by summing all the contributions at different lags [27]. $ns(rs\_preci)$ is a natural cubic spline (ns) of the 21-day rolling sum of precipitation (rs_preci) with 3 df included as another main exposure; $ns(time)$ is a natural cubic spline of day with 7 df per year to control for seasonality and long-term trends. $DOW_t$ is an indicator variable for the day of the week on day $t$, and $Holiday_t$ is an indicator variable for site-specific national holidays on day $t$. Using the same approach, we conducted additional analyses by stratifying into age groups of 0–11 months, 12–23 months, and 24–59 months. We have controlled temperature and precipitation for each other in the same model. The relative risk (RR) for both temperature and precipitation was estimated as the ratio of the risk at the 95th percentile compared to the risk at the 1st percentile of each variable.

In the second stage, we applied a multivariable random-effects meta-analysis to assess heterogeneity in the effect estimates of temperature/precipitation on diarrhoea between sites and subsequently, a meta-regression approach was adopted to identify site-specific indicators that could account for the observed heterogeneity [28]. In specific, we used the meta-regression models to evaluate the association between site-specific indicators and the effect estimates of temperature/precipitation on diarrhoea. Moreover, we considered the precision of the effect estimates obtained from the first stage, as estimated by their covariance, giving less weight to more imprecise estimates. The analysis was conducted separately for each site-specific indicator. Annual mean temperature and annual mean of daily total precipitation also served as meta-predictors in this study, explaining the variations in temperature-diarrhoea and precipitation-diarrhoea associations between different sites. We assessed heterogeneity between sites using the $I^2$ statistic, Cochran Q test, and likelihood ratio (LR) tests for the site-specific indicators. In the context of our study, a hypothesis for the Cochran Q test was tested to assess whether residual heterogeneity exists across different sites. This test determines if variations between sites significantly impact our observed outcomes. Similarly, the LR test assesses whether a more complex model (with an additional predictor) significantly improves the fit compared to a simpler intercept-only model.

## Sensitivity analysis

Several sensitivity analyses were performed to test the robustness of our results. We fitted our models to control for relative humidity and dew point temperature. We changed the lag period

for temperature (14 and 28 days), the duration of the rolling sum for precipitation (14 and 28 days), the df for temperature (4 to 5), and the df for the 21-day rolling sum for precipitation (4 to 5).

All statistical analyses were conducted with R software version 4.2.0 (The R Foundation for Statistical Computing, Vienna, Austria). The *dlnm*, *mixmeta and ncdf4* packages were used for nonlinear model with distributed lags, meta regression and ERA5-land data extraction.

## Results

### Description of diarrhoea and weather factors

Table 1 shows summary statistics for the outcomes by age group and exposure variables in each study site. The analysis included 66,056 ACD cases in total, with the most (n = 21,751, 33%) cases in Pakistan and the fewest (n = 2384, 4%) in Kenya. Diarrhoea was more common among 0–11 months (n = 28,392, 43%) than in 12–23 months (n = 22,487, 34%) and 24–59 months (n = 15,177, 23%). Daily ambient temperatures ranged from 15.2˚C in Bangladesh to 37.7˚C in Mali over the study period. The temperature variability in the locations from South Asia was comparable to that in Sub-Saharan Africa. Furthermore, the temperature distribution was notably narrow in Kenya and the widest in Pakistan (S1 Table). S1 Fig presents the distribution of weather variables to provide a clearer understanding of their seasonal range and the corresponding outcome variables analyzed in the study. Childhood diarrhoea showed clear seasonality in most countries; however, these patterns varied across different geographic locations.

### Association between temperature and diarrhoea

Fig 1 and Table 2 show the cumulative effect of temperature on ACD. The association between temperature and ACD tended to be non-linear by site; although the ACD risk in Mali and Bangladesh leveled off when temperatures exceeded 26˚C. The incidence of diarrhoea increased with rising temperatures in Mozambique (RR [17˚C vs. 29˚C] = 4.96, 95% CI: 2.81–8.76) and Bangladesh (RR [17˚C vs. 30˚C] = 8.07, 95% CI: 3.70–17.60), but decreased in Mali (RR [20˚C vs. 33˚C] = 0.25, 95% CI: 0.11–0.55) and Pakistan (RR [17˚C vs. 31˚C] = 0.24, 95% CI: 0.12–0.51). We plotted the distributed lag for a maximum of 21 days and age-specific ACD for each site (S3A–S3D and S4 Figs). At the 95[th] percentile temperature, the largest effect estimates were observed on the current day (lag 0) in all seven countries, and a positive effect was observed up to 21 days in Mozambique, Kenya, and Bangladesh (S3A Fig). A significant positive association between ACD and temperature (at the 95[th] percentile) was found at lag 0–1 in The Gambia, at lag 4–18 in Mozambique, at lag 10–16 in Kenya, and at lag 4–9 in Bangladesh. A significant negative association of ACD and temperature (at the 95[th] percentile) was found at lag 3, lag 10–15 in Mali, and at lags 2–3 and 15–19 in Pakistan. In Bangladesh, with temperatures above the 95[th] percentile, the cumulative RRs were higher among infants (aged 0–11 months) than in toddlers (aged 12–23 months) and children (aged 24–59 months). Conversely, in Mozambique, RRs were higher in toddlers compared with those in infants and children, although the CIs overlapped among these age groups in both countries (S4 Fig and S2 Table).

### Association between precipitation and diarrhoea

The association between precipitation and ACD exhibited a monotonic increasing pattern in Gambia, Mali, Kenya, and India; like temperature in Mali and Bangladesh also levelled off at higher precipitations. Pakistan also levelled off when the cumulative precipitation over 21 days exceeded 30 mm but in the opposite direction (Fig 2). The incidence of diarrhoea increased

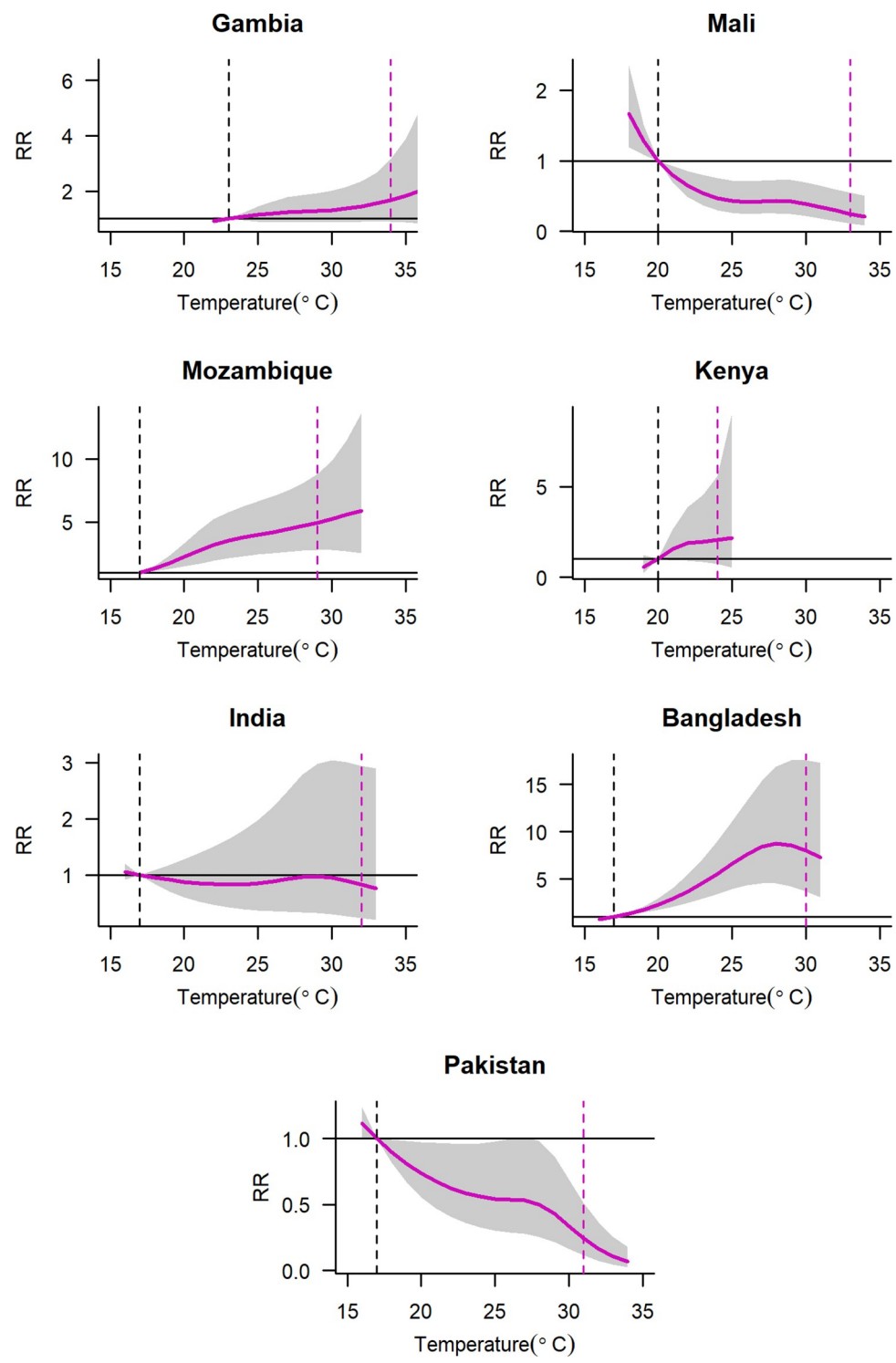

**Fig 1. Association between daily mean temperature over 21 days and all-cause diarrhoea.** Purple solid lines represent point estimates of relative risk (RR); grey-shaded polygons are 95% confidence intervals. The RRs were estimated using a distributed lag non-linear model (DLNM), contrasting the daily mean temperature at various percentiles against the reference temperature. Dashed black vertical lines are reference temperature at the 1st percentile, and dashed purple vertical lines are the temperature at the 95th percentile.

**Table 2. Relative Risk of Temperature and Precipitation in All-Cause Diarrhoea in Seven Study Sites.**

| Study Site | Temperature | | | Precipitation | | |
|---|---|---|---|---|---|---|
| | 1st pctl | 95th pctl | Relative Risk (95% CI) | 1st pctl | 95th pctl | Relative Risk (95% CI) |
| The Gambia | 23 | 34 | 1.67 (0.89, 3.14) | 0 | 535 | 1.36 (0.93, 1.99) |
| Mali | 20 | 33 | **0.25 (0.11, 0.55)** | 0 | 542 | **1.51 (1.03, 2.20)** |
| Mozambique | 17 | 29 | **4.96 (2.81, 8.76)** | 6 | 306 | 0.89 (0.73, 1.08) |
| Kenya | 20 | 24 | 2.05 (0.75, 5.58) | 57 | 550 | 1.13 (0.71, 1.81) |
| India | 17 | 32 | 0.83 (0.23, 2.94) | 0 | 769 | **1.55 (1.08, 2.22)** |
| Bangladesh | 17 | 30 | **8.07 (3.70, 17.60)** | 0 | 1011 | 1.18 (0.87, 1.61) |
| Pakistan | 17 | 31 | **0.24 (0.12, 0.51)** | 0 | 161 | **0.77 (0.67, 0.89)** |

We used a cross-basis function for temperature with a 21-day lag period both smoothed using a natural cubic spline with 3 degrees of freedom. Precipitation is based on the rolling sum of 21 days smooth using a natural cubic spline with 3 degrees of freedom. pctl, percentile; CI, confidence interval.

with precipitation in Mali (RR [0 vs 542 mm] = 1.51, 95% CI 1.03–2.20)) and India (RR [0 vs 769 mm] = 1.55, 95% CI 1.08–2.22). The RR declined significantly (RR [0 vs 161 mm] = 0.77, 95% CI 0.67–0.89) in Pakistan (Table 2). Children aged 12–23 months and 24–59 months in Mali and India, tended to exhibit higher risk than the children aged 0–11 months. However, there were overlapping CIs among these age groups in both countries (S5 Fig and S2 Table). The wide confidence intervals suggest that precipitation-diarrhoea results indicate uncertainty, especially in the sites in Mali, Mozambique, Bangladesh, and Pakistan.

The meta-analysis showed substantial heterogeneity in the associations of temperature–diarrhoea and precipitation–diarrhoea across sites, with an $I^2$ of 84.2% and 67.5%, respectively. Results of meta-regression showed that pathogens (bacteria) and dehydration treated with ORS could modify the temperature–diarrhoea association across different study sites. Only nutritional status (stunting) was found to be an important predictor that modified the precipitation–diarrhoea association (Table 3).

## Sensitivity analysis

In sensitivity analyses, the estimates for the association of temperature and precipitation with diarrhoea were generally similar when adjusting for possible time-varying covariates (relative humidity and dew point). The RRs were not substantially changed by changing the lag days, number of rolling sum days, or changing the exposure df; only some estimated ranges of CIs were reduced or increased (S6 and S7 Figs).

## Discussion

We found varying patterns of diarrhoea morbidity in relation to temperature and precipitation across sites in this study. Specifically, we observed an independent increase in diarrhoea morbidity associated with high temperatures in Mozambique and Bangladesh. In contrast, we found a significant inverse association between temperature and diarrhoea incidence in Mali and Pakistan. Regarding precipitation, the results demonstrated that high precipitation was independently associated with increased diarrhoea in Mali and India whereas a negative association was found in Pakistan.

The relationship between temperature and ACD varied among countries. In Mozambique and Bangladesh, the evidence indicated that high temperatures contributed to ACD morbidity. This was consistent with the findings of previous single-site studies [9,11,12,18,29–31]. A study in Mozambique found that each 1˚C increase on the hottest day of the concurrent week was associated with a 3.64% increase in the incidence of diarrhoea [14]. Another study in

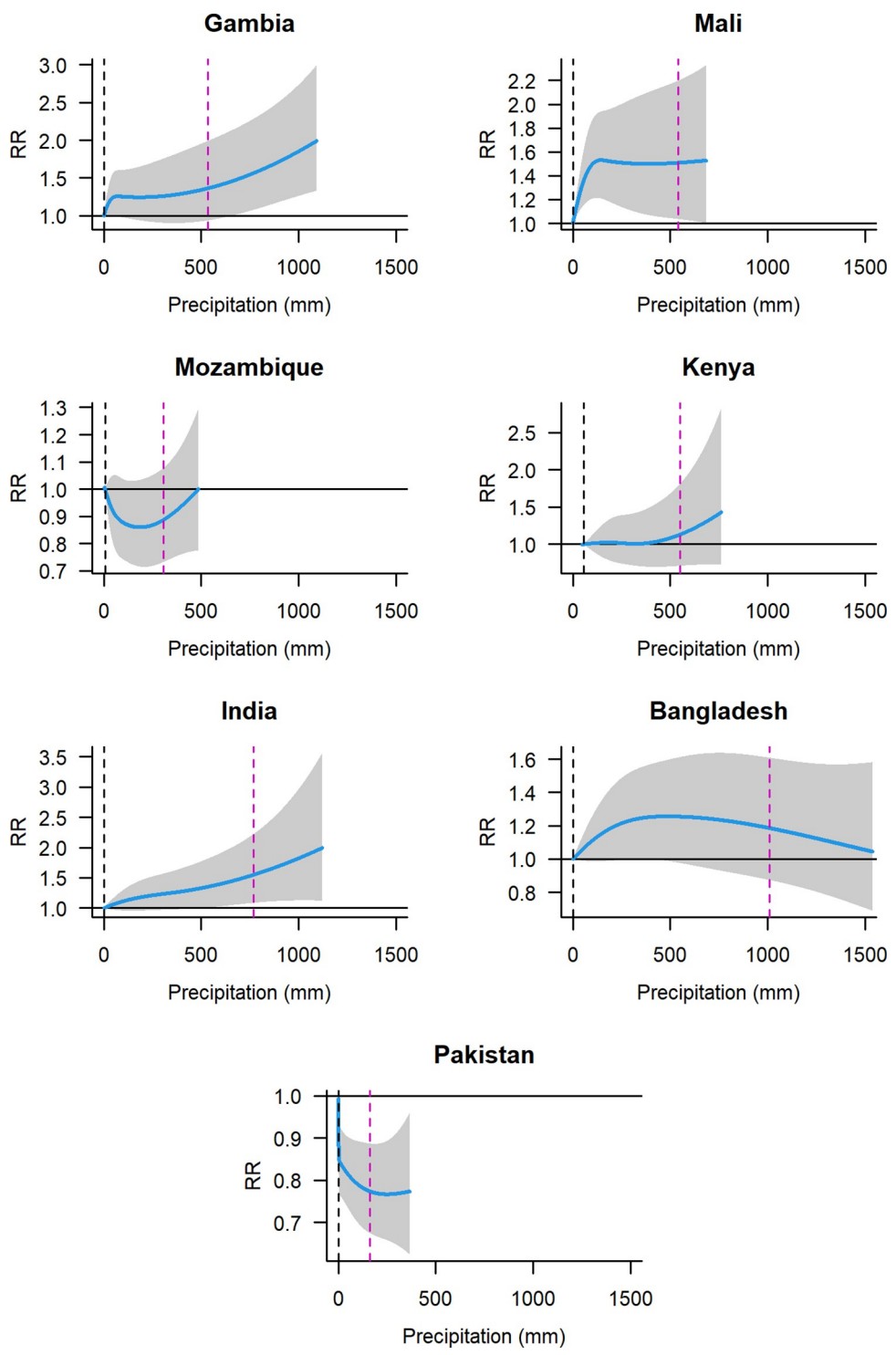

**Fig 2. Association between precipitation over 21 days and all-cause diarrhoea.** Blue solid lines represent point estimates of relative risk (RR); grey-shaded polygons are 95% confidence intervals. The RRs were estimated using a natural cubic spline function, contrasting the three weeks of moving sum precipitation at various percentiles against the reference precipitation. Dashed black vertical lines are reference precipitation at the 1st percentile, and dashed purple vertical lines are precipitation at the 95th percentile.

**Table 3. Results of meta-regression model for temperature–diarrhoea and precipitation–diarrhoea associations.**

| | Temperature | | | | Precipitation | | | |
|---|---|---|---|---|---|---|---|---|
| | Coefficient* | $I^2$ (%) | Q test (p-value) | LR test (p-value) | Coefficient* | $I2$ (%) | Q test (p-value) | LR test (p-value) |
| Intercept only | 2.581 | 84.2 | <0.001 | | 1.199 | **67**.5 | <0.001 | |
| Annual average temperature (˚C) | -0.428 | 82.9 | <0.001 | 0.511 | 0.063 | 60.0 | <0.001 | 0.498 |
| Annual average precipitation (mm) | 0.291 | 82.6 | <0.001 | 0.304 | 0.024 | 57.2 | 0.002 | 0.571 |
| Gross domestic product (per capita) | 0.001 | 81.3 | <0.001 | 0.131 | 0.000 | 58.8 | 0.002 | 0.597 |
| Population density (per sq km) | 0.001 | 86.5 | <0.001 | 0.821 | 0.000 | 71.1 | <0.001 | 0.939 |
| Poverty level | -0.010 | 86.4 | <0.001 | 0.894 | -0.005 | 72.1 | <0.001 | 0.945 |
| Urbanization (mean score) | -8.772 | 80.3 | <0.001 | 0.122 | 1.148 | 65.9 | <0.001 | 0.16 |
| Treated water (%) | -0.060 | 84.9 | <0.001 | 0.829 | -0.002 | 64.1 | <0.001 | 0.308 |
| Drinking water (tube well) (%) | 0.065 | 81.0 | <0.001 | 0.073 | 0.000 | 68.0 | <0.001 | 0.389 |
| Hand washing (%) | 0.074 | 85.6 | <0.001 | 0.6 | -0.005 | 71.9 | <0.001 | 0.691 |
| Stunting (%) | -0.063 | 85.4 | <0.001 | 0.296 | -0.027 | 26.1 | 0.016 | **0.015** |
| Viruses (%) | -0.086 | 86.6 | <0.001 | 0.904 | -0.006 | 59.7 | <0.001 | 0.095 |
| Bacteria (%) | 0.133 | 82.3 | <0.001 | **0.008** | -0.012 | 63.9 | <0.001 | 0.373 |
| Protozoa (%) | -0.160 | 85.5 | <0.001 | 0.339 | 0.001 | 72.5 | <0.001 | 0.92 |
| Moderate-severe dehydration (%) | -0.024 | 86.1 | <0.001 | 0.772 | 0.005 | 55.9 | 0.003 | 0.391 |
| Oral rehydration solution (%) | 0.062 | 84.1 | <0.001 | **0.024** | -0.006 | 56.6 | 0.002 | 0.076 |

*The coefficient represents the association between each meta-predictor and the RR (temperature–diarrhoea and precipitation–diarrhoea associations). RR is estimated as the risk ratio at the 95[th] percentile compared to the risk at the 1[st] percentile. Heterogeneity ($I^2$), Cochran Q test was tested to assess whether residual heterogeneity exists across different sites, and LR test assesses whether a more complex model (with an additional predictor) significantly improves the fit compared to a simpler intercept-only model.

Dhaka, Bangladesh found that an average 1˚C increase in temperature over the previous 4 weeks was associated with a 6.5% increased incidence of non-cholera and non-rotavirus diarrhoea [19]. However, it is important to note that the direction of this association varies by region. In Mali and Pakistan, we found that diarrhoea morbidity decreased with increasing temperature. A few epidemiological studies have suggested that the risk of infectious diarrhoea decreases with a rise in temperature; a time-series study conducted by Thiam et al. in 2017 showed that nighttime temperatures above 26˚C were negatively associated with under-five diarrhoea incidence [32], Wang et al. reported an association between meteorological factors and diarrhoea incidence in southern China for 11 years of time-series studies and found a high overall cumulative RR for low mean temperature [33]. Another study found in Botswana a negative association between temperature and diarrhoea during the wet season [34]. The results of multivariable meta-regression showed that some modifiers could explain heterogeneity among seven sites in seven countries. We found that the prevalence of bacterial pathogens modified the association between high temperature and child diarrhoea. A systematic review and meta-analysis study showed significant differences of association with temperature among bacterial pathogens [35]. The majority of studies reported positive associations between temperature and bacterial infections [36] and a study showed a negative association [37]. We also found that ORS used at home for treating diarrhoea had a significantly different effect on the association between temperature and diarrhoea. Prompt and appropriate use of ORS can mitigate the severity and duration of diarrhoeal episodes, regardless of the surrounding temperature conditions. In fact, ORS can help patients recover at home without needing to go to the hospital in many cases. Studies have shown contrasting patterns of ORS adoption in Kenya and Mexico [38]. These modifiers reflected the differences in the local epidemiology and health

behaviours of the seven sites in seven countries. These results indicate that temperature may have varying impacts on diarrhoeal diseases in different settings.

There was a monotonic increase in precipitation-related diarrhoea in three locations, but this varied in direction across other locations. A steady rise in ACD cases was observed in The Gambia, Kenya, and India, with Mali and India having significant results of 542 mm and 769 mm of precipitation over a 3-week moving sum. In contrast, the plot for Bangladesh suggests that the effect of precipitation on diarrhea weakens under higher precipitation values. It should be mentioned that wide intervals indicate uncertainty and make it challenging to draw strong conclusions about the relationship between precipitation and child diarrhoea. However, our findings are consistent with those of previous studies conducted in different regions [24,39,40]. For example, Carlton et al. performed a study in Ecuador and found that heavy rainfall events were associated with increased diarrhoea incidence following dry periods and a decrease following wet periods [15]. A study in India showed that heavy precipitation can cause water contamination and disrupt sanitation systems, with a breakdown of infrastructure [29]. Additionally, heavy precipitation is associated with an increased risk of diarrhoea through its impact on river levels [41], causing river levels to rise, which can lead to flooding. Flooding can lead to contamination of water sources with sewage and other pollutants, making it difficult to access clean water and increasing the risk of waterborne illnesses such as diarrhoea. Heavy precipitation can also cause a flushing action that can send pathogens, including those that cause diarrhoea, into surface and groundwater sources. This can happen when heavy precipitation causes surface runoff, which can carry bacterial and protozoan pathogens from contaminated areas [42] such as from overflowing septic tanks, latrines, or sewage treatment plants, into nearby water sources [43]. In contrast, our analysis revealed an inverse relationship between precipitation and diarrhoea in Pakistan, which receives the lowest amount of rain among the countries in our study (see Table 1). This implies that low precipitation levels increase the risk of diarrhoea in children; previous studies support these findings [17,31]. A lack of water owing to drought has been associated with outbreaks of several waterborne diseases. For instance, in places where water is scarce, there has been an increased occurrence of infections like cholera, *Escherichia coli* infection, and leptospirosis [44], which could be attributed to poor hygiene practices and the consumption of contaminated water with higher pathogen concentrations [7]. However, the association between precipitation and diarrhoea differed across different sites in our study. We investigated this heterogeneity and found that nutritional status (stunting) could modify this association. It is less clear how to interpret these variations. A study investigated the association between rainfall variations and child diarrhoea in 15 Sub-Saharan African countries and shows that height for age z score is no longer significant by climate zones [8]. Nevertheless, a study showed watery diarrhoea was 63% less likely to cause death in children with acute malnutrition than those with better nutritional status [45]. Previous studies have indicated that combined water, sanitation, and handwashing (WASH) interventions had no additive benefit over single interventions in reducing diarrhea prevalence among children under five. However, single sanitation and handwashing interventions resulted in significant reductions, almost similar to the effects of combined water, sanitation, and handwashing (WASH) and combined WASH and nutrition interventions in rural Bangladesh [46]. This might explain the weak evidence we observed for the association between relative risk with treated water, drinking water, and handwashing. A trial study design also found that the effect of combined WASH interventions was concentrated in the monsoon season and following periods of heavy rainfall, whereas water and handwashing interventions were less influenced by weather [47]. Future studies should consider combined WASH-related variables to gain a more comprehensive understanding of their effects.

Our age-stratified analysis revealed that children under two years of age were more susceptible to diarrheal diseases in relation to weather factors. This could be explained by the fact that very young children may stay indoors with their mothers, who are often engaged in domestic activities that expose them to contaminated water and food sources [48]. The immune systems of young children may not be fully developed and may not be able to cope with the increased pathogen load that results from climate change [49]. Children under two are often exploring their environment, putting objects in their mouths, and may have less developed hygiene habits [50]. This increases their exposure to pathogens that can cause diarrhoeal diseases. Moreover, when they start to walk, they may encounter more environmental exposure, like temperature and precipitations, could increase their risk of infection [51].

The study period concluded in 2011, and weather patterns in the study locations may have significantly changed. These changes could influence the present-day applicability of our findings, particularly in regions where climate variability plays a significant role in environmental processes. While our results provide valuable insights into present-day conditions, future research could explore how these findings generalize to current conditions by incorporating more recent data and examining shifts in climate and weather patterns in the study location. The relationship between temperature, precipitation, and diarrhoea is complex and can vary depending on the specific pathogens involved. Different pathogens have different transmission mechanisms and survival rates under different environmental conditions. For example, some diarrhoeal pathogens may survive longer in warmer temperatures whereas others may thrive in wetter conditions. Therefore, conducting pathogen-specific analysis in future studies is crucial to better understand the relationship between environmental factors and diarrhoeal diseases.

## Strengths and limitations of this study

Several studies have investigated the association between diarrhoea and environmental factors, such as temperature and precipitation, within a single country or multiple sites within one country. However, there is little research on this association across multiple countries using consistent methodologies. We addressed this knowledge gap by analysing data from multiple sites and geographically diverse countries using well-tested methods to investigate the association between diarrhoea and environmental factors, such as temperature and precipitation. The daily dataset enabled the application of advanced DLNM to assess the lag patterns, particularly in the context of temperature.

The seven study sites represent a range of child health indicators in developing countries as well as urban, peri-urban, and rural settings. Our study's stratified analysis by age is a rare approach in similar research, providing more nuanced insights into how different age groups are affected. To our knowledge, this is the largest study addressing the independent impact of temperature and precipitation on ACD, with consideration of non-linear and delayed dependencies.

Some limitations of our study must be acknowledged. First, we only considered one site in each country; therefore, our findings may not be nationally representative. Second, the study period was relatively short, and the CIs of the estimates were wide for some sites, indicating variability and uncertainty in the results. Third, while we employed a time series model to explore the relationship between child diarrhoea with temperature and precipitation, conducting season-specific site-wise time series modelling is challenging due to the limited statistical power resulting from our small sample size. Lastly, pathogens causing diarrhoea infections might behave differently under varying climate conditions, and our dataset could not fully support the development of the pathogen-specific regression model due to unequal

probabilities of daily sampling of cases with unknown weights. Future research should focus on the mechanisms by which temperature and precipitation influence diarrhoeal infectious diseases. This includes pathogen-specific analysis and projection of future risks of diarrhoea considering different socioeconomic pathways.

## Conclusion

Our study highlights the essential role of ambient temperature and precipitation in influencing diarrhoeal infections in South Asian and Sub-Saharan regions. Specifically, we observed varying susceptibility to diarrheal diseases related to high temperatures in different sites: Manhica, Mozambique, and Mirzapur, Bangladesh showed increased infections, while Bamako, Mali, and Karachi, Pakistan exhibited decreased incidence. Similarly, high precipitation was linked to increased diarrhoeal infections in Bamako, Mali and Kolkata, India, but decreased infections in Karachi, Pakistan. These variations may be explained by factors such as pathogen types, the use of ORS for treating dehydration, and nutritional status.

## Supporting information

**S1 Table. Summary statistics information of ERA5 land.**
(DOCX)

**S2 Table. Summary of age-specific relative risk of temperature (˚C) and precipitation over 21 days on all-cause diarrhoea for seven countries at 95[th] percentile temperature and precipitation.**
(DOCX)

**S3 Table. List of site-specific meta predictors.**
(DOCX)

**S1 Fig. Time-series plots of the daily number of all-cause diarrhoea cases, moderate to severe diarrhoea cases, daily mean temperature, and precipitation between 2008 and 2011 in the study countries.**
(DOCX)

**S2 Fig. Distribution of ERA5 daily temperature and precipitation.**
(DOCX)

**S3 Fig. Relationships between temperature and ACD specific to 95[th] percentile temperature, lag 7, 14 and 21 for each site.**
(DOCX)

**S4 Fig. Association between daily mean temperature over 21 days and all-cause diarrhoea stratified by age group.**
(DOCX)

**S5 Fig. Association between daily precipitation over 21 days and all-cause diarrhoea stratified by age group.**
(DOCX)

**S6 Fig. Cumulative relative risks (RRs) with 95% Cis [95[th] vs. 1[st] Pctl] by site in sensitivity analyses for the Temperature-Diarrhoea model.**
(DOCX)

**S7 Fig. Cumulative relative risks (RRs) with 95% Cis [95th vs. 1st] by site in sensitivity analyses for the Precipitation-Diarrhoea model.**
(DOCX)

## Acknowledgments

We gratefully acknowledge the support of the University of Maryland, which is the PI Institute of GEMS study. We thank Toshihiko Sunahara, PhD, and Michiko Toizumi, PhD for editing a draft of this manuscript.

## Author Contributions

**Conceptualization:** Nasif Hossain, Masahiro Hashizume.

**Data curation:** Nasif Hossain.

**Formal analysis:** Nasif Hossain.

**Funding acquisition:** Nasif Hossain, Masahiro Hashizume.

**Investigation:** Nasif Hossain.

**Methodology:** Nasif Hossain, Lina Madaniyazi, Chris Fook Sheng Ng, Masahiro Hashizume.

**Project administration:** Masahiro Hashizume.

**Resources:** Xerxes Tesoro Seposo, Paul L. C. Chua, Rui Pan, Abu Syed Golam Faruque.

**Software:** Masahiro Hashizume.

**Supervision:** Lina Madaniyazi, Chris Fook Sheng Ng, Dilruba Nasrin, Masahiro Hashizume.

**Validation:** Nasif Hossain.

**Writing – original draft:** Nasif Hossain.

**Writing – review & editing:** Nasif Hossain, Lina Madaniyazi, Chris Fook Sheng Ng, Dilruba Nasrin, Xerxes Tesoro Seposo, Paul L. C. Chua, Rui Pan, Abu Syed Golam Faruque, Masahiro Hashizume.

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
