## [Decision Letter · Decision Letter 0]

15 Feb 2024

Dear Dr. Hossain,

Thank you very much for submitting your manuscript "Short-term associations of diarrhoeal diseases in children with temperature and precipitation in seven low- and middle-income countries from Sub-Saharan Africa and South Asia in the Global Enteric Multicenter Study" for consideration at PLOS Neglected Tropical Diseases. As with all papers reviewed by the journal, your manuscript was reviewed by members of the editorial board and by several independent reviewers. In light of the reviews (below this email), we would like to invite the resubmission of a significantly-revised version that takes into account the reviewers' comments. 

We think all the reviewers' comments are reasonable. Please address or rebut each one in your point-by-point response to reviewer comments.

We cannot make any decision about publication until we have seen the revised manuscript and your response to the reviewers' comments. Your revised manuscript is also likely to be sent to reviewers for further evaluation.

Sincerely,

Josh M Colston, Ph.D.

Academic Editor

Justin Remais

Section Editor

We think all the reviewers' comments are reasonable. Please address or rebut each one in the authors' response.

Reviewer's Responses to Questions

**Key Review Criteria Required for Acceptance?**

**Methods**

-Are the objectives of the study clearly articulated with a clear testable hypothesis stated?

-Is the study design appropriate to address the stated objectives?

-Is the population clearly described and appropriate for the hypothesis being tested?

-Is the sample size sufficient to ensure adequate power to address the hypothesis being tested?

-Were correct statistical analysis used to support conclusions?

-Are there concerns about ethical or regulatory requirements being met?

Reviewer #1: The statistical analysis section can be improved by providing a general explanation of how the cross-basis function of temperature and its lags work since this approach for dealing with nonlinearity and lag is quite complex . A citation for this approach might be useful for those interested in looking further. 

This is a small number of sites for a meta regression, especially with many covariates. Some discussion of this could be helpful. Based on the current explanation in methods, I'm having trouble understanding the columns in Table 3 of results and the explanation that certain covariates were important predictors for modifying either temperature-diarrhoea or precipiation diarrhoea association. Were the additional variables only included as interactions with temp and precip or am I misunderstanding the meta-regression approach? Please elaborate in methods by explaining what hypotheses are being tested with both the LR test and Q test so this is clear.

Reviewer #2: The methods used in this study are appropriate, comprehensive and well-articulated. I just have some questions:

-What was the rationale behind the selection of a 21-day rolling sum of daily total precipitation, as mentioned in lines 116-118. While the authors have appropriately cited three relevant references utilizing a similar lag period, it would be valuable to explicitly articulate the reasons for this choice and offer insights into the rationale for other lag times.

-The authors mention the utilization of a time series regression with a quasi-Poisson distribution. Could the authors elaborate on their approach to addressing overdispersion? Did they employ a robust sandwich estimator for standard errors? Additionally, if the authors used Generalized Additive Models for the spline fits, kindly specify this in the methods section.

-Combining WASH-related variables, specifically treated water and handwashing, to evaluate their combined impact on the association between temperature or precipitation and ACD might provide valuable insights. A study found that combined WASH had more reductions in diarrhea prevalence than water intervention alone (https://linkinghub.elsevier.com/retrieve/pii/S2214109X17304904).

-Consideration of urbanicity/rurality of the study sites could be valuable. I suggest exploring the addition of these variables in the meta-regression or perhaps something to be discussed in the Discussion section. 

-Further clarification is needed on the meta-analysis methodology. Specifically, could the authors confirm whether a two-stage hierarchical approach, as they referenced Sera and Gasparrini et al. 2022, was adopted? If so, please explicitly outline this in the methods section.

Reviewer #3: - For clarity, the authors should state early in the methods section that the GEMS study used a case-control design. Though other publications have reported the design of the GEMS study, it is difficult to follow this manuscript without more details about the design. For example, there are no details about how controls were sampled and what the underlying target population was in each site. Further, to contextualize the study, more details are needed on whether the sites were primary urban or rural. 

- The methods should justify why the investigators considered a rolling 21-day sum for precipitation to be appropriate to capture any pathogen transport following weather events and the incubation period for the pathogens that primarily cause diarrhea in these settings. For temperature, the statistical analysis section alludes to the use of a 21-day lag, but this is not clearly justified either. 

- There are insufficient details about the data sources for covariates such as GDP, poverty level, use of treated water, nutritional status, etc. Were all of these collected as part of the GEMS study? How were these measured? It is difficult to assess the risk of bias without further details. 

- As noted below, it would be valuable to investigate relationships with heavy rainfall and extreme heat. It might also be helpful to investigate relationships within different seasons.

**Results**

-Does the analysis presented match the analysis plan?

-Are the results clearly and completely presented?

-Are the figures (Tables, Images) of sufficient quality for clarity?

Reviewer #1: I hope my comments about the methods can help clear up how to interpret Table 3.

Reviewer #2: -The figures presented in the results section are visually appealing. Optional: The authors may try having the same scales and range for the x- and y-axes in Fig 1 and 2 for each study site for easier comparison. 

-Regarding lines 177-178, where the authors suggest a linear association between temperature and ACD in most countries. Based on Fig 1 and Fig S1, it seems that the relationships are non-linear. Could the authors clarify?

-The non-significant findings of all wealth and economic variables in the meta-regression is a bit surprising since previous studies found association between socioeconomic position/status and child diarrhea. I'm wondering if the authors could explore more granular measures of wealth, such as an asset-based wealth index or maternal education in the analysis?

-Additionally, including descriptive statistics of the variables included in the meta-regression in Table 1, along with asset-based wealth index/maternal education and urbanicity/rurality, if applicable, would enhance the contextual understanding of each study site.

-Clarification is needed whenever the authors refer to "independent temperature/precipitation." Does this imply that these exposures were modeled separately or controlled for each other?

Reviewer #3: - The site-specific precipitation results have such wide confidence intervals that it is difficult to say much about the slope of the relationship in most of the countries (Mali, Mozambique, Bangladesh, Pakistan). The results presentation and discussion section should be modified to account for this.

**Conclusions**

-Are the conclusions supported by the data presented?

-Are the limitations of analysis clearly described?

-Do the authors discuss how these data can be helpful to advance our understanding of the topic under study?

-Is public health relevance addressed?

Reviewer #1: The conclusions are presented well.

Reviewer #2: -Line 339 mentions "ambient temperature..." Could the authors clarify the term "ambient"? In my understanding, it could refer to indoor temperature, and this may affect how people might interpret the conclusion. 

-Also the authors acknowledged in their limitation that the study sites might not be nationally-representative.It might be beneficial to address this consideration in the conclusion when referring to the countries, for example, by specifying "certain areas/regions in Bangladesh."

Reviewer #3: - The discussion should not draw conclusions about the impact of climate change on diarrhea (e.g., line 301) but rather should focus on the association between weather and diarrhea risk. 

- The authors state that investigating pathogen-specific relationships with weather will be crucial, but they did not do so even though they appeared to have the data for it. Was the reason not to look at bacterial diarrhea, viral diarrhea, etc. because of the limited sample size? 

- The factors identified in the meta-regression (ORS use, nutritional status) are not clearly linked to weather. It is not clear if the authors hypothesize that weather changes these risk factors or if they are just strongly associated with diarrhea-related healthcare utilization.

**Editorial and Data Presentation Modifications?**

Reviewer #1: (No Response)

Reviewer #2: -The authors may consider using "multivariable" meta-analysis/meta-regression instead of "multivariate" since they assessed only one outcome with multiple predictors.

Reviewer #3: - In the abstract, it is not clear if the RRs are for a one-unit increase in temperature/precipitation. 

- On line 302, regarding the point about young children being exposed to contaminated water and food in the home, it seems just as likely that older children are exposed to more pathogens because they have exposures both inside and outside the home.

**Summary and General Comments**

Reviewer #1: This is a well thought out analysis of both nonlinear and lagged effects of temperature and precipitation on daily all-cause diarrhea using GEMS data. There have been studies that look at nonlinear relationships between temperature and precipitation, their lags, and various diarrheal outcomes, though perhaps as you say, not independent impact of temperature and precipitation on ACD, with consideration of non-linear and delayed dependencies. The analysis uses an interesting approach of cubic splines and cross-basis functions in order to allow for nonlinearity and assessment of lags through relative risk with 95% CIs but the methods could be elaborated to allow the general reader (and even statisticians) to understand the approach a little better. The same goes for the meta-regression and the hypothesis tests being conducted. Overall, this is a nice extension to some work related to seasonality and diarrheal pathogens with a unique approach for showing nonlinearity and lag.

Reviewer #2: I appreciate the opportunity to review this manuscript. The paper is very well-written with interesting findings.

Reviewer #3: This manuscript presents associations between temperature and precipitation and diarrhea using data from the GEMS study. The results differ substantially between sites, with some sites showing increased risk of diarrhea with increasing temperature and some showing decreased risk. The strength of relationships in opposite directions in different sites is somewhat surprising. It seems that the authors could do more to use the rich GEMS dataset to understand the heterogeneity in these findings. For example, they did not fully leverage pathogen data to understand how temperature and precipitation are related to bacterial vs. viral diarrhea. They also did not investigate relationships with indicators for heavy rainfall or extreme heat. In addition, unlike prior studies, they did not investigate whether the precipitation findings differed when restricting to rainfall following dry periods. Most prior studies have found that diarrhea risk is higher following heavy rainfall if the preceding period was dry (concentration dilution hypothesis). The authors seemed to have the data to investigate this but did not do so.

PLOS authors have the option to publish the peer review history of their article (what does this mean?). If published, this will include your full peer review and any attached files.

Reviewer #1: Yes: Ben J. Brintz

Reviewer #2: No

Reviewer #3: No
---

## [Decision Letter · Decision Letter 1]

12 Jun 2024

Dear Dr. Hossain,

Thank you very much for submitting your manuscript "Short-term associations of diarrhoeal diseases in children with temperature and precipitation in seven low- and middle-income countries from Sub-Saharan Africa and South Asia in the Global Enteric Multicenter Study" for consideration at PLOS Neglected Tropical Diseases. As with all papers reviewed by the journal, your manuscript was reviewed by members of the editorial board and by several independent reviewers. The reviewers appreciated the attention to an important topic. Based on the reviews, we are likely to accept this manuscript for publication, providing that you modify the manuscript according to the review recommendations. 

Thank you for thoroughly addressing the reviewer comments. The manuscript is much improved. We hope you can respond to some last comments, specifically Reviewer 3 comments regarding how the findings and models are reported.

Sincerely,

Alexandra K Heaney

Academic Editor

Justin Remais

Section Editor

Thank you for thoroughly addressing the reviewer comments. The manuscript is much improved. We hope you can respond to some last comments, specifically Reviewer 3 comments regarding how the findings and models are reported.

Reviewer's Responses to Questions

**Key Review Criteria Required for Acceptance?**

**Methods**

-Are the objectives of the study clearly articulated with a clear testable hypothesis stated?

-Is the study design appropriate to address the stated objectives?

-Is the population clearly described and appropriate for the hypothesis being tested?

-Is the sample size sufficient to ensure adequate power to address the hypothesis being tested?

-Were correct statistical analysis used to support conclusions?

-Are there concerns about ethical or regulatory requirements being met?

Reviewer #1: See my comments in results. I think the critique could be mostly addressed in methods.

Reviewer #2: The authors have answered my questions regarding the Methods and made the necessary changes based on my comments.

Reviewer #3: 1. The authors use an appropriate data source for assessing meteorological variables, with the right time and space granularity needed for their analysis. 

2. Please state which temperature measurement (outdoor vs dew point) was used in the primary analysis.

3. Lines 133-134: “Although ‘rolling sum’ may seem technical, they refer to the same concept regarding delayed effect.”

o I don’t fully agree with this claim. While the rolling sum captures precipitation over a 21-day lookback period, it’s not the same as what’s happening with the distributed lags for temperature. For example, the 21-day lag temperature may describe the effect of temperature 21 days prior to the diarrhea event, independent of temperatures in subsequent days. The 21-day rolling sum of precipitation aggregates rainfall across the whole period, making it difficult to separate the impacts of precipitation within that time. In the most extreme case, the total precipitation might have been entirely concentrated on the same day as a diarrhea event and would not represent a delayed effect. If the authors would like to capture a delayed effect in precipitation, they could add a cross-basis function with lagged total precipitation (perhaps summed over a small window). 

4. Lines 157-159: “The overall cumulative relative risk (RR) for both temperature and precipitation was estimated as the ratio of the risk at the 95th percentile compared to the risk at the 1st percentile of each variable.“

o Based on Figures 1 and 2, it looks like cumulative RRs were also calculated for all other percentiles? This should be explicitly stated in the methods. Given that these are primary results, additional detail is also needed on how cumulative relative risks are estimated. It seems like they aggregate over all lagged temperatures, but how is this done? This information is needed to guide the interpretation of the results and should not solely rely on an external citation.

**Results**

-Does the analysis presented match the analysis plan?

-Are the results clearly and completely presented?

-Are the figures (Tables, Images) of sufficient quality for clarity?

Reviewer #1: The meta-analysis results (Table 3) are still a little difficult to follow. Though the additional explanation of the hypothesis tests in methods was helpful, I think the authors could be very clear that the metaanalyis is done on the estimate (as relative risk?) for the association between Temp and Diarrhea or Prec. and Diarrhea and the predictors are accounting for additional variance. After understanding that, it's clear how you might additionally adjust for temp and precipitation but it's not stated in methods. It's also not clear if the metaregression/analysis is done on patient-level data or site-level data. Finally, would it be possible to include effect estimates of the predictors in addition to the I-squared value and p-values?

Reviewer #2: I have no further comments. Thank you to the authors for making the changes accordingly.

Reviewer #3: 1. The abstract reports relative risks for diarrhea associated with 1 unit increases in temperature and precipitation. However, the reported results seem to correspond to the cumulative RRs that estimated differences in risk between the 1st vs 95th percentile values (as reported in Table 2). This statement also implies a strictly linear relationship between diarrhea and temperature + precipitation, but the authors have only assessed these relationships with non-linear models.

2. Generally, cumulative RRs don’t have an intuitive interpretation for how temperature + precipitation impact diarrhea risk. The authors should be more specific in how they describe these estimates and avoid conflating it with the risk associated with a simple one-unit increase in temperature + precipitation. 

o It may be more clear to report and plot RRs for fixed lags and/or fixed temperature + precipitation values

3. Throughout the results, the authors state how RR increases or decreases with weather but they seem to be describing trends in incidence. 

o I would recommend making the edits similar to below:

Original [Lines 203-205]: The RR of diarrhoea increased in Mozambique (4.96, 95% confidence interval [CI] 2.81-8.76) 

Edited: The incidence of diarrhoea increased with temperature in Mozambique (RR [XX vs YY degrees C] = 4.96, 95% confidence interval [CI] 2.81-8.76)

**Conclusions**

-Are the conclusions supported by the data presented?

-Are the limitations of analysis clearly described?

-Do the authors discuss how these data can be helpful to advance our understanding of the topic under study?

-Is public health relevance addressed?

Reviewer #1: No new comments

Reviewer #2: I have no further comments.

Reviewer #3: 1. It would help to provide a brief description of the climate of each site, particularly in reference to the timing and severity of rainy seasons. This would also emphasize the climate diversity of sites selected for this study. 

2. If the study only included diarrhea cases from hospitalized individuals, it would be important for authors to discuss how the exclusion of non-hospitalized diarrhea cases influences their results. 

3. Lines 330-334: Previous studies have indicated that combined water, sanitation, and handwashing (WASH) interventions led to greater reductions in diarrhoea prevalence compared to single water, sanitation, or handwashing interventions. This might explain the weak evidence we observed for the association between relative risk with treated water, drinking water, and handwashing. 

o A secondary analysis of this trial also found that the effect of combined WASH + sanitation interventions was concentrated in the monsoon season and following periods of heavy rainfall, but that water and handwashing interventions were less influenced by weather (https://ehp.niehs.nih.gov/doi/10.1289/EHP13807)

**Editorial and Data Presentation Modifications?**

Reviewer #1: No new comments

Reviewer #2: (No Response)

Reviewer #3: 1. In the text of results, the authors describe trends shown in the figures to convey the direction of the weather-diarrhea relationships. (Lines 219-221: “The association between precipitation and ACD infection exhibited a monotonic increasing pattern … levelled off at higher precipitations… (Fig 2)”). However, it’s hard to see these trends in Figures 1 + 2 since all facets are plotted with the same y-axis scale. The authors should consider allowing the y-axis of the plots to vary by country or include more informative descriptions in the text.

**Summary and General Comments**

Reviewer #1: Overall I think the revised manuscript looks great. The strengths and perhaps novelty of the study include the interesting dlnm approach for assessing associations between weather variables and diarrhea which includes an exploration of temperature effects over 21 lag days. The meta analysis section is quite brief in the paper and as a result, the results are slightly hard to follow and interpret. I think a little more in-depth explanation of the metaanalysis and regression could be included in methods for clarity.

Reviewer #2: I would like to thank the authors for the changes made in the manuscript. I think this is a very interesting and timely paper. I only have a minor comment/ suggestion before the acceptance of the manuscript which is regarding lines 346-351 (of the draft with track changes). I would like to clarify that in the cited article (https://linkinghub.elsevier.com/retrieve/pii/S2214109X17304904), there was no significant reduction in diarrhea prevalence for single water intervention. But for single sanitation and handwashing interventions, they were significant reductions which were almost similar with the effects of combined WSH and combined WSH + nutrition in rural Bangladesh.

Reviewer #3: Thank you for the opportunity to review this paper. This work addresses an important research gap and demonstrates how associations between weather and diarrhea varies across different settings. The authors successfully integrated data from GEMS and ERA-5, as well as various data products capturing socioeconomic factors, to complete a comprehensive assessment of factors that may influence these relationships. The inclusion of GDP, poverty, population density, and other health indicators was unique and provide a strong foundation for future studies that will investigate these relationships. However, there are some inconsistencies in how the methods and results were reported throughout the manuscript. These discrepancies should be resolved prior to publication, to allow for more clear interpretation of these important results.

PLOS authors have the option to publish the peer review history of their article (what does this mean?). If published, this will include your full peer review and any attached files.

Reviewer #1: Yes: Ben Brintz

Reviewer #2: No

Reviewer #3: No

Figure Files:

Data Requirements:

Reproducibility:

References

---

## [Decision Letter · Decision Letter 2]

19 Sep 2024

Dear Dr. Hossain,

We are pleased to inform you that your manuscript 'Short-term associations of diarrhoeal diseases in children with temperature and precipitation in seven low- and middle-income countries from Sub-Saharan Africa and South Asia in the Global Enteric Multicenter Study' has been provisionally accepted for publication in PLOS Neglected Tropical Diseases.

Before your manuscript can be formally accepted you will need to complete some formatting changes and changes to Figures in response to final comments of Reviewer 3, which you will receive in a follow up email. A member of our team will be in touch with a set of requests.

Best regards,

Alexandra K Heaney

Academic Editor

Justin Remais

Section Editor

Reviewer's Responses to Questions

**Key Review Criteria Required for Acceptance?**

**Methods**

-Are the objectives of the study clearly articulated with a clear testable hypothesis stated?

-Is the study design appropriate to address the stated objectives?

-Is the population clearly described and appropriate for the hypothesis being tested?

-Is the sample size sufficient to ensure adequate power to address the hypothesis being tested?

-Were correct statistical analysis used to support conclusions?

-Are there concerns about ethical or regulatory requirements being met?

Reviewer #1: The methods are much clearer now.

Reviewer #2: (No Response)

Reviewer #3: The reviewers have satisfactorily addressed my prior comments.

**Results**

-Does the analysis presented match the analysis plan?

-Are the results clearly and completely presented?

-Are the figures (Tables, Images) of sufficient quality for clarity?

Reviewer #1: (No Response)

Reviewer #2: (No Response)

Reviewer #3: 1. Currently, the text of the results only references Figure S1 to discuss the seasonality of diarrhea prevalence, but that figure also contains really nice visualizations of how temperature/precipitation vary over time in each region. It might be worth explicitly stating that these weather distributions are included in Figure S1 for readers to get a better sense of the range + seasonality of weather variables in the study.

2. I appreciate that the authors were receptive of my suggestion to format the presentation of RRs to include the exact temperature values of the 1st and 99th percentiles. This provided value context to the results that helps make the results more interpretable.

**Conclusions**

-Are the conclusions supported by the data presented?

-Are the limitations of analysis clearly described?

-Do the authors discuss how these data can be helpful to advance our understanding of the topic under study?

-Is public health relevance addressed?

Reviewer #1: (No Response)

Reviewer #2: (No Response)

Reviewer #3: 1. The authors state that there was a “monotonic increase in precipitation-related diarrhoea” (line 326), but this does not appear to be supported by the plots in Figure 2. For example, the plot for Mali suggests that increases to precipitation past ~100mm don’t affect the RR of diarrhea and the plot of Bangladesh suggests that the effect of precipitation on diarrhea weakens under higher precipitation values. The authors used appropriate language in discussing these findings in their results section and may consider editing their discussion to make it clearer that the monotonic increases weren’t observed in all study sites.

2. The study period ended in 2011 and it’s likely that the climate and weather patterns in study sites have changed since then. The authors might consider adding discussion on the generalizability of their results to present day conditions.

**Editorial and Data Presentation Modifications?**

Reviewer #1: (No Response)

Reviewer #2: (No Response)

Reviewer #3: 1. For Figures 1 and 2, it would help to add details to the captions on what contrasts were used to estimate the RRs.

**Summary and General Comments**

Reviewer #1: This revision of the manuscript is much improved with more details in both the methods and results regarding the meta-regression.

Reviewer #2: The authors have made the recommended changes. I have no further comments. Well done!

Reviewer #3: 1. I thank the authors for their revisions to this manuscript; their edits have provided clarity to their statistical methods and helped with the interpretation of these impactful results. This work is a valuable contribution to the growing literature on the influence of weather on childhood diarrhea. I have only minor comments that could be addressed prior to publication.

PLOS authors have the option to publish the peer review history of their article (what does this mean?). If published, this will include your full peer review and any attached files.

Reviewer #1: **Yes: **Ben Brintz

Reviewer #2: No

Reviewer #3: No

---

## [Editor Report · Acceptance letter]

27 Sep 2024

Dear Dr. Hossain,

We are delighted to inform you that your manuscript, "Short-term associations of diarrhoeal diseases in children with temperature and precipitation in seven low- and middle-income countries from Sub-Saharan Africa and South Asia in the Global Enteric Multicenter Study," has been formally accepted for publication in PLOS Neglected Tropical Diseases.

Best regards,

Shaden Kamhawi

co-Editor-in-Chief

Paul Brindley

co-Editor-in-Chief
